# Mapping the molecular motions of 5-HT₃ serotonin-gated channel by voltage-clamp fluorometry

**Laurie Peverini\*, Sophie Shi, Karima Medjebeur, Pierre-Jean Corringer\***

Institut Pasteur, Université Paris Cité, CNRS UMR 3571, Channel-Receptors Unit, Paris, France

\*For correspondence:
laurie_peverini@hotmail.com
(LP);
pjcorrin@pasteur.fr (P-JeanC)

**Competing interest:** The authors declare that no competing interests exist.

**Abstract** The serotonin-gated ion channel (5-HT₃R) mediates excitatory neuronal communication in the gut and the brain. It is the target for setrons, a class of competitive antagonists widely used as antiemetics, and is involved in several neurological diseases. Cryo-electron microscopy (cryo-EM) of the 5-HT₃R in complex with serotonin or setrons revealed that the protein has access to a wide conformational landscape. However, assigning known high-resolution structures to actual states contributing to the physiological response remains a challenge. In the present study, we used voltage-clamp fluorometry (VCF) to measure simultaneously, for 5-HT₃R expressed at a cell membrane, conformational changes by fluorescence and channel opening by electrophysiology. Four positions identified by mutational screening report motions around and outside the serotonin-binding site through incorporation of cysteine-tethered rhodamine dyes with or without a nearby quenching tryptophan. VCF recordings show that the 5-HT₃R has access to four families of conformations endowed with distinct fluorescence signatures: 'resting-like' without ligand, 'inhibited-like' with setrons, 'pre-active-like' with partial agonists, and 'active-like' (open channel) with partial and strong agonists. Data are remarkably consistent with cryo-EM structures, the fluorescence partners matching respectively apo, setron-bound, 5-HT bound-closed, and 5-HT-bound-open conformations. Data show that strong agonists promote a concerted motion of all fluorescently labeled sensors during activation, while partial agonists, especially when loss-of-function mutations are engineered, stabilize both active and pre-active conformations. In conclusion, VCF, though the monitoring of electrophysiologically silent conformational changes, illuminates allosteric mechanisms contributing to signal transduction and their differential regulation by important classes of physiological and clinical effectors.

## eLife assessment

This **valuable** study applies voltage clamp fluorometry to provide new information about the function of serotonin-gated ion channels 5-HT3AR. The authors **convincingly** investigate structural changes inside and outside the orthosteric site elicited by agonists, partial agonists, and antagonists, helping to annotate existing cryo-EM structures. This work confirms that the activation of 5-HT3 receptors is similar to other members of this well-studied receptor superfamily. The work will be of interest to scientists working on channel biophysics but also drug development targeting ligand-gated ion channels.

## Introduction

Pentameric ligand-gated ion channels (pLGICs), comprising nicotinic acetylcholine (nAChRs), serotonin (5-HT₃Rs), glycine (GlyRs), and gamma-amino butyric acid (GABA_ARs) receptors, are dynamic

transmembrane proteins that mediate neuronal and non-neuronal communication (*Nemecz et al., 2016*). Among them, 5-HT$_3$Rs are cation-selective excitatory channels expressed in central and peripheral nervous systems, notably in the gut and gut-brain axis (*Gibbs and Chakrapani, 2021*). 5-HT$_3$Rs are encoded by five different genes (*HTR3A, HTR3B, HTR3C, HTR3C, HTR3E*), giving rise to five different subunits (A to E) that assemble as homo- or hetero-pentamers. A class of synthetic competitive antagonists called setrons target these receptors and are widely used in the clinic as antiemetics, notably to counteract vomiting and nausea induced by chemotherapy and radiotherapy. 5-HT$_3$Rs are also potential targets to treat irritable bowel syndrome and are implicated in neurological diseases such as schizophrenia, Parkinson's disease, and depression. There is therefore much interest surrounding the molecular mechanisms governing 5-HT$_3$Rs function and pharmacological regulation (*Mawe and Hoffman, 2013*; *Fakhfouri et al., 2019*; *Tsitsipa et al., 2022*).

A hallmark of pLGICs, and among them 5-HT$_3$R, is that their action is mediated by varying population equilibrium between allosteric states that are differentially stabilized by the binding of ligands (*Monod et al., 1965*; *Corradi et al., 2009*). Agonists promote the allosteric transition between resting (closed channel) and active (open channel) states, while their prolonged application promotes a slower transition to desensitized (agonist-bound-closed channel) conformations.

The first high-resolution structure of mouse homo-pentameric 5-HT$_{3A}$R was solved by X-ray crystallography in complex with stabilizing nanobodies, followed by numerous structures solved by cryo-electron microscopy (cryo-EM) without ligands or in complex with the agonist 5-HT, a series of setron competitive antagonists and an allosteric modulator (*Hassaine et al., 2014*; *Polovinkin et al., 2018*; *Basak et al., 2018b*; *Basak et al., 2018a*; *Basak et al., 2019*; *Basak et al., 2020*; *Zarkadas et al., 2020*; *Zhang et al., 2021*). The 5-HT$_{3A}$R displays a typical pLGIC structure, where each subunit consists of a β-sandwich connected with loops composing the extracellular domain (ECD), four α-helices composing the transmembrane domain (TMD) and an intracellular domain. Agonists and competitive antagonists bind at the ECD interface between two subunits, at the orthosteric site composed of the so-called loops A, B, and C from the principal subunit and D, E, and F from the complementary subunit, with loop C bridging the subunit interface. In the presence of detergent C12E9 with or without added lipids, or in saposin nanodiscs, solved structures in the absence of orthosteric ligand (apo) were assigned as resting-like (*Hassaine et al., 2014*; *Basak et al., 2018b*; *Zhang et al., 2021*). Those in complex with setrons significantly diverged from the resting-like class and were called inhibited states (*Polovinkin et al., 2018*; *Basak et al., 2019*; *Basak et al., 2020*; *Zarkadas et al., 2020*). The structures captured with bound 5-HT all feature rather similar motions of the ECD, with 5-HT binding promoting a closure (in a capping motion) of loop C around the agonist, a global compaction, and a tilt of each subunit ECD. This generates a marked reorganization of the ECD-TMD interface, including in some cases an outward motion of the M2-M3 loop. However, the structures strongly diverge in the TMD, some featuring an apparently open pore and others a non-conducting pore (*Polovinkin et al., 2018*; *Basak et al., 2018a*). In addition, reconstitution into saposin-based nanodiscs captured distinct open conformations with symmetrical or asymmetrical conformation of the TMD (*Zhang et al., 2021*). Therefore, these atomic structures led to a complex landscape of more than the three putative conformations (resting, active, desensitized). Furthermore, 5-HT$_3$R's wide conformational landscape accessibility seems to be governed by both orthosteric effectors and the lipids and/or detergents surrounding the TMD. Assigning these high-resolution structures to actual physiological states contributing to the electrophysiological response at the cell membrane remains to be established (*Howard, 2021*).

Complementary techniques that follow the structural reorganizations of the cell-expressed receptors are thus needed to identify structural motions contributing to gating (i.e. the pathway between resting and activated/open states) and to help annotate high-resolution structures to physiologically relevant conformational states (*Howard, 2021*). Voltage-clamp fluorometry (VCF), which monitors receptors expressed at the *Xenopus* oocyte plasma membrane, is well suited for this aim. VCF performs simultaneous measurement of channel opening by two-electrode voltage-clamp electrophysiology (TEVC) and of local protein motions by variation of fluorescence from covalently attached probes (*Mannuzzu et al., 1996*). This technique has been applied to various pLGICs, notably to human 5-HT$_{3A}$R with the generation of three sensors surrounding the orthosteric site as well as one sensor on the mouse 5-HT$_{3A}$R located at the extracellular entrance of the pore (*Polovinkin et al., 2018*; *Munro et al., 2019*).

In the present study, we have generated four fluorescent sensors grafted at new locations on the mouse 5-HT$_{3A}$R. We characterized their phenotypes following the binding of various agonists, partial agonists, and clinically relevant antagonists, with or without introduction of loss-of-function mutations. Simultaneous recordings of electrical currents and variations of fluorescence revealed different local and global reorganizations of the ECD depending on the pharmacological conditions. By following the conformational reorganizations leading to both electrophysiologically silent and active conformations, VCF data identify intermediate conformations within or outside the path of 5-HT$_3$R activation. These data provide unique structural information on membrane-embedded proteins to annotate known high-resolution structures to physiologically relevant allosteric states.

## Results
### Generation of four fluorescent sensors along the ECD

To guide the design of fluorescent sensors, we inspected the regions undergoing the largest reorganizations in the various cryo-EM structures, mainly mapping the subunit interface from the apex of the ECD to the ECD-TMD interface and the upper part of the TMD. We introduced cysteines by mutagenesis and covalently labeled them with the fluorescent probe MTS-TAMRA, allowing conjugation of a rhodamine dye with the protein main chain through a flexible 6-atom linker (CH$_2$-S-S-CH$_2$-NH-CO). Since rhodamine fluorescence is sensitive to its local molecular environment, this generates conformational sensors around the graft position (*Munro et al., 2019*). In order to increase the intensity of fluorescent variations and to report more precisely the motion observed, we also implemented the tryptophan-induced quenching method (*Mansoor et al., 2010*; *Figure 1A*). In this technique, the cysteine-labeled rhodamine is introduced together with a nearby tryptophan since the indole moiety of the Trp side chain robustly quenches rhodamine fluorescence when in Van der Waals contact. Such pairs sense the distance between the fluorophore and indole attachment points and/or local environment, and often amplify the extent of the fluorescence variation when allosteric motions are associated with structural reorganization at this level.

Nineteen TAMRA-labeled cysteine mutants were screened by VCF under perfusion of high concentration of 5-HT (*Figure 1A*). We verified that the mouse wild-type 5-HT$_3$R (m5-HT$_3$R), that does not carry any single cysteine within the ECD, yielded robust 5-HT-elicited currents upon treatment with MTS-TAMRA but no changes in fluorescence intensity. The screening allowed for the selection of four mutants (one cysteine and three cysteine/tryptophan pairs) endowed with robust variations of both current and fluorescence.

We first identified the single mutation S204C, which is located near the tip of loop C, facing the adjacent subunit and the ECD interface above the orthosteric site (*Figure 1B*). Second, three fluorophore/quencher (cysteine/tryptophan) pairs were selected. In each case, the measured 5-HT-elicited variation of fluorescence (ΔF) was much smaller or unmeasurable when the cysteine but not the tryptophan was introduced, indicating that the dequenching or quenching observed is dominated by Van der Waals interaction between the rhodamine dye and the indole of the tryptophan (*Figure 1—figure supplement 1C*). The pairs are the following:

1. I160C/Y207W: I160C is located in close proximity to the orthosteric site but outside the pocket behind the base of loop C, while Y207W lies inside the pocket nearby bound ligands in agonist and antagonist co-structures. Of note, the aromatic side chain of Y207 contributes to the binding and gating in 5-HT$_3$R, with a strong effect of alanine and serine mutations, but comparatively weaker effect of phenylalanine or unnatural aromatic mutations (*Beene et al., 2004*). The pair is located away from the subunit interface thus reporting tertiary reorganizations (*Figure 1C*).
2. V106C/L131W: V106C is facing the vestibule of the ECD, a large water-accessible channel with several constrictions and ring of charges that lies above the TMD ion channel. L131W is positioned more profoundly in each ECD, its side chain remaining visually accessible through the vestibule. Given the proximity of all subunits in this confined space, the observed fluorescence variation can potentially arise from quenching by tryptophan of any of the five subunits (*Figure 1D*). Of note, we verified that the observed fluorescent quenching is not caused by direct collisional quenching with the hydroxy indole group of perfused 5-HT, that arises at concentrations above 200 μM (see an example of direct quenching by 5-HT perfused at 1000 μM, *Figure 1—figure supplement 1C*).

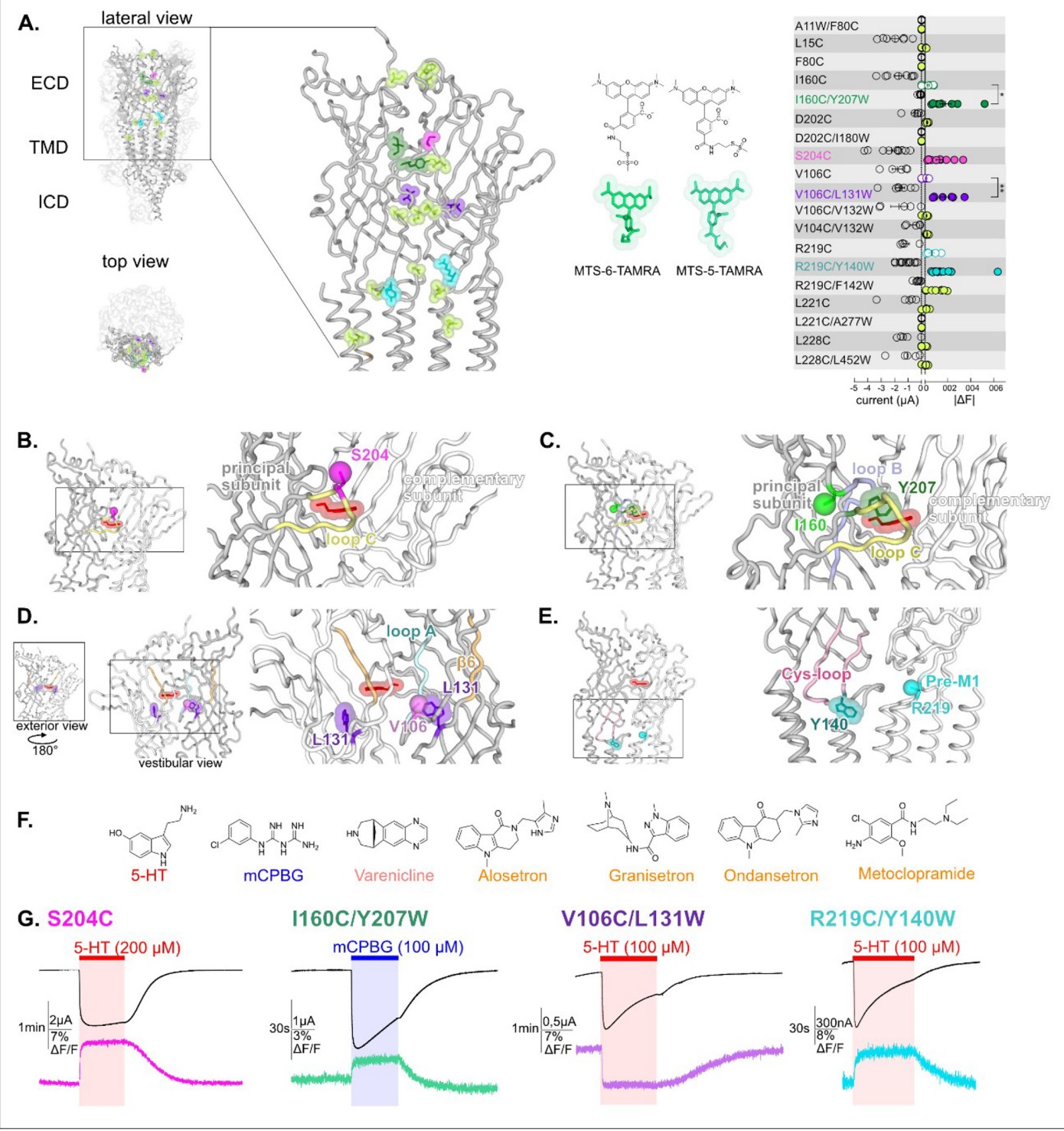

**Figure 1.** Screening of residues located along the extracellular domain (ECD) of m5-HT₃AR as potential anchors for fluorescent probes and establishment of four reporter sensors. (**A**) Left panel: visualization of the residues mutated, TAMRA-labeled and tested for variation of current and fluorescence on the apo cryo-electron microscopy (cryo-EM) structure of m5-HT₃A (from **Basak et al., 2018a**), in lateral (zoom on the ECD) and top view with two subunits in cartoon representation and the three others represented in surface mode (PDB: 6BE1). Middle panel: representation of the two isomers of the labeling fluorescent probe mixture used in this study, MTS-TAMRA (5(6)-carboxytetramethylrhodamine methanethiosulfonate). Right panel: representation of currents evoked by 50–100 µM of 5-HT on the mutants and of the absolute values of variation of fluorescence (difference between the baseline fluorescence and steady-state fluorescence upon 5-HT perfusion) recorded simultaneously (error bars are +/- SEM, n = 5 to 13). Note the representation of the four selected sensors: in dark green, I160C/Y207W; in magenta, S204C; in purple, V106C/L131W, and in cyan,

*Figure 1 continued on next page*

*Figure 1 continued*

R219C/Y140W. (**B**) The sensor S204C, shown on 5-HT-bound conformation (5-HT represented in red; PDB: 6DG8), is located on loop C. Note that the fluorescence variation is significantly increased by the addition of the tryptophan in the sensors I160C/Y207W and V106C/L131W (unpaired t-tests, I160C/Y207W versus I160C p-value = 0.013 (*); V106C/L131W versus V106C p-value = 0,0081 (**)) (**C**) The sensor I160C/Y207W where I160C is the point of labeling with MTS-TAMRA and is represented in ball representation in fluorescent green, Y207 is mutated into tryptophan and colored in dark green. (**D**) The sensor V106C/L131W where V106C is represented in ball representation in purple and L131 is mutated into tryptophan and colored in purple. Note the vestibular positioning of this sensor. (**E**) The sensor R219C/Y140W, where R219C, located in pre-M1 loop, is represented in ball representation in cyan and Y140, located in the Cys-loop, is mutated into tryptophan, and colored in dark cyan. (**F**) Molecular structures of the ligands used in this study: agonists (in red, 5-HT; in blue, *m*-chlorophenylbiguanide [mCPBG]; in salmon, varenicline) and antagonists (all represented in orange, A: alosetron, G: granisetron, O: ondansetron, M: metoclopramide). (**G**) Effect of desensitization on the dynamic of the fluorescence recordings. Examples of desensitizing currents promoted by prolonged perfusion of strong agonists (to elicit robust desensitization): mCPBG perfused on the sensor I160C/Y207W or 5-HT perfused on the sensors S204C, V106C/L131W, and R219C/Y140W. Traces show that the fluorescent signal remains stable during desensitization for all the sensors. Note that the differences in the desensitization kinetics of the displayed traces are due to the variability of different oocyte batches, since the four conditions do not show significant differences in desensitization kinetics (*Figure 1—figure supplement 2*, *Figure 1— source data 1*).

The online version of this article includes the following source data and figure supplement(s) for figure 1:

**Source data 1.** Source data containing fluorescence values for *Figure 1A* right panel.

**Figure supplement 1.** Controls.

**Figure supplement 1—source data 1.** Electrophysiological recordings of wild-type construct upon application of the different agonists and antagonists stated in this study and electrophysiological measurements of the different mutants tested in this study.

**Figure supplement 2.** Desensitization properties.

**Figure supplement 2—source data 1.** Tau values and calculated percent of desensitization for the constructs mentioned in this study.

3. R219C/Y140W are positioned at the ECD/TMD interface. R219C is located in the pre-M1 area and Y140W in the highly conserved Cys-loop. Both positions flank the M2-M3 loop that undergoes a major outward motion in some agonist-bound structures (*Figure 1E*).

## The four sensors monitor fast motions not related to desensitization

The four sensors were recorded on a custom VCF chamber where the ligands are perfused only on the portion of the oocyte from which the fluorescence emission is collected (*Shi et al., 2023*). This ensures that the same population of receptors is recorded in current and fluorescence simultaneously. For all sensors, perfusion of a high concentration of strong agonists (5-HT or *m*-chlorophenylbiguanide [mCPBG], selected depending on the particular sensor, see dedicated section to each sensor) elicit currents reaching maximal value in a few seconds, followed by desensitization appearing with much slower kinetics (*Figure 1G*). To measure the desensitization kinetics, we performed parallel measurements on a dedicated TEVC setup equipped with a fast perfusion system allowing solution exchange in less than a hundred milliseconds (*Gielen et al., 2020*). This shows that all sensors display comparable desensitization kinetics to that of the wild-type receptor (WT), evaluated in the 50–150 s range through mono-exponential fitting (*Figure 1—figure supplement 2*). Thus, activation and desensitization appear well separated in time in the VCF setup. This allows a reasonable evaluation of the extent of activation by measuring the amplitude of the peak current. Concentration-response curves measured at this peak current show that, for all sensors, labeling by MTS-TAMRA has no significant effect in terms of $EC_{50}$ current ($EC_{50}c$) (*Figure 1—figure supplement 1B*).

In fluorescence, labeled S204C, I160C/Y207W, and R219C/Y140W show robust agonist-elicited dequenching and V106C/L131W agonist-elicited quenching. In all cases, the rise time of the fluorescence variations (ΔFs) are in the same range to that of the rise time of the currents (1–10 s range depending on the particular sensor and the agonist concentration, *Supplementary file 1*). In contrast, prolonged applications of agonist show no significant variation of the fluorescence during the desensitization phase (with the fluorescent signal remaining stable), providing evidence that the sensors do not report movements related to desensitization (*Figure 1G*). As far as the four sensors are concerned, VCF data suggest that the ECD (and its labels) does not undergo notable conformational changes between activated and desensitized states.

## Competitive antagonists elicit agonist-like reorganizations at the orthosteric site that do not spread to the lower part of the ECD

We first explored the action of competitive antagonists on the various sensors. To this end, we applied to the same oocyte saturating concentrations of the agonists 5-HT, mCPBG, and var (varenicline) as well as a selection of four competitive antagonists of different molecular structure: alosetron, granisetron, ondansetron, and metoclopramide (*Kilpatrick et al., 1990*; *Lummis et al., 2011*; *Thompson and Lummis, 2006*). We used a 3 µM concentration of each antagonist, which is far above their nanomolar binding affinities measured on the WT receptor.

On the two sensors neighboring the orthosteric site, antagonists produce no current, but robust ΔFs are observed in the same direction as agonists do (*Figure 2A and C*). In addition, the amplitude of ΔFs evoked by antagonists are in the same range than that of agonists for granisetron and ondansetron while it is significantly higher for alosetron and lower for metoclopramide. This suggests that, locally, antagonists elicit similar reorganizations as agonists do but with various amplitudes (*Figure 2A and C*, left panels).

In contrast, the sensors at the vestibular site (V106C/L131W) and at the ECD-TMD interface (R219C/Y140W) show no antagonist-elicited current nor ΔF, suggesting that the conformational effects elicited by these competitive antagonists do not spread to the vestibule and lower part of the ECD (*Figure 3A and C*, left panels).

## Correlation between the orthosteric site motions and ion channel activation

The sensor S204C displays electrophysiological properties identical to that of the WT, both in terms of agonist potency and efficacy. $EC_{50}c$ are identical between S204C and WT, i.e., 1.8 and 1.8 µM for 5-HT, 1.4 and 1.1 µM for mCPBG, and 20.6 and 21.1 µM for var, respectively (*Table 1*). mCPBG appears as a near full agonist, eliciting 89% and 92% of maximal 5-HT currents on S204C and WT, respectively, while var appears as a partial agonist eliciting only 33% and 39% of maximal 5-HT currents on S204C and WT, respectively (*Table 2*). All data on WT are consistent with TEVC data reported in the literature (*Lummis et al., 2011*).

The strong agonists 5-HT and mCPBG elicit similar maximum fluorescence variation (ΔFmax) at saturation. Concentration-response curves show that the variations of current (ΔIs) and fluorescence are well correlated with similar $EC_{50}$ fluorescence ($EC_{50}f$) and $EC_{50}c$. This shows that agonists promote local reorganizations around the binding site that are correlated to the opening of the channel, suggesting a concerted motion of the orthosteric site and the ion channel during activation. As for strong agonists, var displays similar $EC_{50}f$ and $EC_{50}c$. However, it displays a ΔFmax similar to that of strong agonists despite activating only 33% of the current. Therefore, var at saturation promotes full reorganization of the binding site suggesting that among the population displaying a variation of fluorescence, only a fraction shows an open channel (*Figure 2A*).

To further investigate the S204C sensor on a different allosteric background, we introduced the strong loss-of function mutation N101K. Substitution of N101 by several natural or unnatural amino acids indeed produces a marked increase in agonist $EC_{50}$ (*Price et al., 2008*; *Sullivan et al., 2006*). N101 is located just below the orthosteric site on loop A and has been proposed to contribute to the allosteric network coupling the orthosteric site to the ion channel. In combination with S204C, N101K displays, as expected, a loss-of-function phenotype characterized by: (1) a 31-fold and 9-fold increase in $EC_{50}c$ for 5-HT and mCPBG, respectively, (2) a decreasing efficacy of 5-HT as compared to mCPBG, as already reported, and (3) a decreasing efficacy of var yielding no detectable current (*Figure 2B* and *Tables 1 and 2*). Interestingly, the effects on the fluorescence variations were comparatively weaker, N101K causing only a 4-fold and a 1.4-fold increase in $EC_{50}f$ for 5-HT and mCPBG, respectively, while var causes a ΔF higher than 5-HT at saturation (173%, *Figure 2B*, left panel and *Table 2*) with a 1.6-fold increase in $EC_{50}f$ (*Table 1* and *Figure 2B*). Therefore, N101K on S204C decorrelates the ΔF and ΔI. For 5-HT and mCPBG, ΔFs are already robust at agonists concentrations not eliciting sizable currents (for instance at 5 µM 5-HT and 1 µM mCPBG, see *Figure 2B*). For var, it elicits quantitative ΔF but no current.

Finally, the sensor I160C/Y207W displays by itself a marked loss-of-function phenotype characterized by: (1) a 7-fold and 17-fold increase in $EC_{50}c$ for mCPBG and 5-HT as compared to the WT, respectively (*Table 1*) and (2) at saturation, mCPBG is the most efficient agonist, while 5-HT elicits

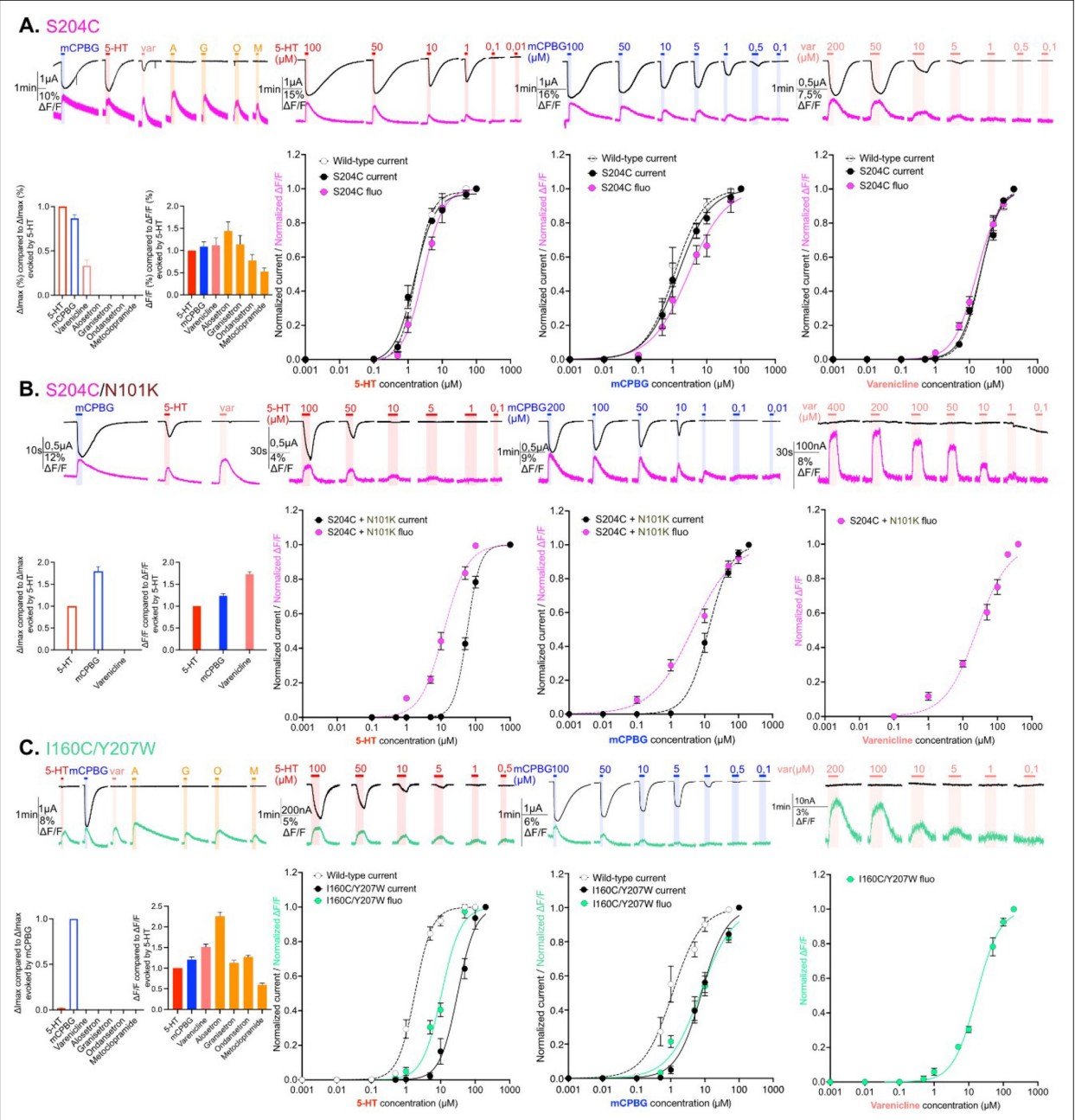

**Figure 2.** Electrophysiological and fluorescence characterization of S204C and I160C/Y207W, two sensors located in proximity to the ligand-binding site. (**A**) Exploration and characterization of the sensor S204C. Left upper panel: macroscopic ligand-gated currents (in black) and fluorescence (in magenta) recorded at –60 mV on S204C labeled with MTS-TAMRA evoked by saturating concentrations of agonists (in red, 5-HT, 200 µM; in blue, *m*-chlorophenylbiguanide [mCPBG], 200 µM; in salmon, varenicline, 400 µM) and antagonists (all represented in orange, A: alosetron, G: granisetron, O: ondansetron, M: metoclopramide, all at 3 µM). Left bottom panel: graphical representation of ligand-induced relative changes of current and fluorescence compared to 5-HT. The normalized values for all the ligands are compared with the mean values obtained for 5-HT. Middle left top panel: representative recording of current and fluorescence variations of S204C labeled with MTS-TAMRA upon different concentrations of perfused 5-HT. Middle left bottom panel: dose-response curves for ΔI (black) and ΔF (magenta) with mean and SEM (normalized by the maximum current of each oocytes) for application of 5-HT. Middle right top panel: representative recording of current and fluorescence variations of S204C labeled with MTS-TAMRA upon different concentrations of perfused mCPBG. Middle right bottom panel: dose-response curves for ΔI (black) and ΔF (magenta) for application of mCPBG. Right top panel: representative recording of current and fluorescence variations of S204C labeled with MTS-TAMRA upon different concentrations of perfused varenicline. Right bottom panel: dose-response curves for ΔI (black) and ΔF (magenta) for application of varenicline. (**B**) Effect of loss-of-function mutation N101K on the sensor S204C. Same experiments and legends as for panel A, but the construct here is the sensor S204C with the additional loss-of-function mutation N101K. (**C**) Exploration and characterization of the sensor I160C/Y207W. Same experiments and

*Figure 2 continued on next page*

*Figure 2 continued*

legends as for panel A, but here currents are represented in black and fluorescence in green. Note also that for the left lower panel, currents and fluorescence have been compared to mCPBG instead of 5-HT (panels A and B, *Figure 2—source data 1*; panel C, *Figure 2—source data 2*). For the entire figure, n are at least 5 and error bars are represented as +/- SEM.

The online version of this article includes the following source data for figure 2:

**Source data 1.** Electrophysiological and fluorescence measurements of the constructs explored in *Figure 2* -1.

**Source data 2.** Electrophysiological and fluorescence measurements of the constructs explored in *Figure 2* -2.

only 2% of its currents and var fails to activate the channel (*Table 2*, *Figure 2C*, left panel). Of note, this phenotype is caused by both I160C and Y207W mutations, that individually cause a 17-fold and 12-fold increase in the $EC_{50}c$ for 5-HT, respectively (*Figure 1—figure supplement 1A*, right panel). Given the position of these mutations within and around the orthosteric site, their effect is likely due to a mix of alteration of both gating and binding affinity of the agonists. The phenotype of I160C/Y207W resembles that of S204C/N101K, with a leftward shift of the $\Delta F$ over the $\Delta I$ curve for the partial agonist 5-HT, and robust $\Delta F$ but no $\Delta I$ for var (*Figure 2C*).

Altogether, VCF data obtained for these two sensors close to the binding site highlight three families of conformations, the apo conformations, the agonist-bound active conformations characterized by a concerted reorganization of the orthosteric site and the ion channel (opening), and ligand-bound intermediate conformations characterized by reorganization of the orthosteric site with a closed channel. On the S204C that displays a WT-like phenotype, these intermediate conformations are partially populated for the partial agonist var. On the two loss-of-function backgrounds (S204C/N101K and I160C/Y207W), these intermediates are fully stabilized by var that behaves as an antagonist and are partially populated for 5-HT and mCPBG at sub-saturating concentrations.

## Correlation between vestibular and ECD-TMD interface motions and ion channel activation

VCF recordings of both V106C/L131W and R219C/Y140W sensors show reproducible currents upon mCPBG perfusion at saturation but not in the case of long recordings required for dose-response curves, where they were non-reproducible. In contrast, 5-HT and var elicit robust and reproducible currents. For both sensors, quantification of the desensitization kinetics on the fast perfusion TEVC setup suggest a slight, although non-significant, increase in desensitization kinetics for 5-HT and var but shows surprisingly very fast desensitization kinetics for mCPBG (*Figure 1—figure supplement 2*.). This explains why, in the VCF setup endowed with a relatively slow perfusion system, the very transient mCPBG-elicited activation peak is truncated by desensitization, precluding its further analysis.

V106C/L131W and R219C/Y140W both display a gain-of-function phenotype that is moderate for the former (1.5-fold and 2.7-fold decrease $EC_{50}c$ for 5-HT and var, respectively), and more pronounced for the latter (5.5-fold and 22.5-fold decrease $EC_{50}c$ for 5-HT and var, respectively). For both sensors, var appears as a partial agonist eliciting around 30% of the 5-HT currents (*Figure 3A and C*; *Table 2*). We also combined R219C/Y140W with the loss-of function N101K mutation. N101K reverts the R219C/Y140W gain-of-function phenotype, the triple mutant R219C/Y140W/N101K displaying $EC_{50}c$ of 5-HT and var close to that of the WT (*Figure 3B*).

These three sensors show a good correlation between $EC_{50}c$ and $EC_{50}f$ for the strong agonist 5-HT, suggesting that the motions reported by fluorescence at the vestibule and ECD-TMD interface are concerted with channel opening. For the partial agonist var on V106C/L131W and R219C/Y140W, the $\Delta Fmax$ (40–50% of that of 5-HT) is comparable to the $\Delta Imax$ (30–40% of that of 5-HT), suggesting in the first approximation that var at saturation promotes the same transition as 5-HT, but only partially. However, concentration-response curves on V106C/L131W show a small yet significant decorrelation of fluorescence and current, suggesting the occurrence of intermediate conformations. R219C/Y140W/N101K confirms this idea with var. On this construct, var acts as a strongly partial agonist eliciting at saturation only 3% of the 5-HT currents, while it displays 37% of the $\Delta F$ evoked by 5-HT (*Figure 3B*, left panel). In addition, var shows a leftward shift of the fluorescence dose-response curve as compared to the current dose-response curve (with $EC_{50}c$ being 15.6 μM and $EC_{50}f$ is 5.2 μM) (*Figure 2B*, right panel).

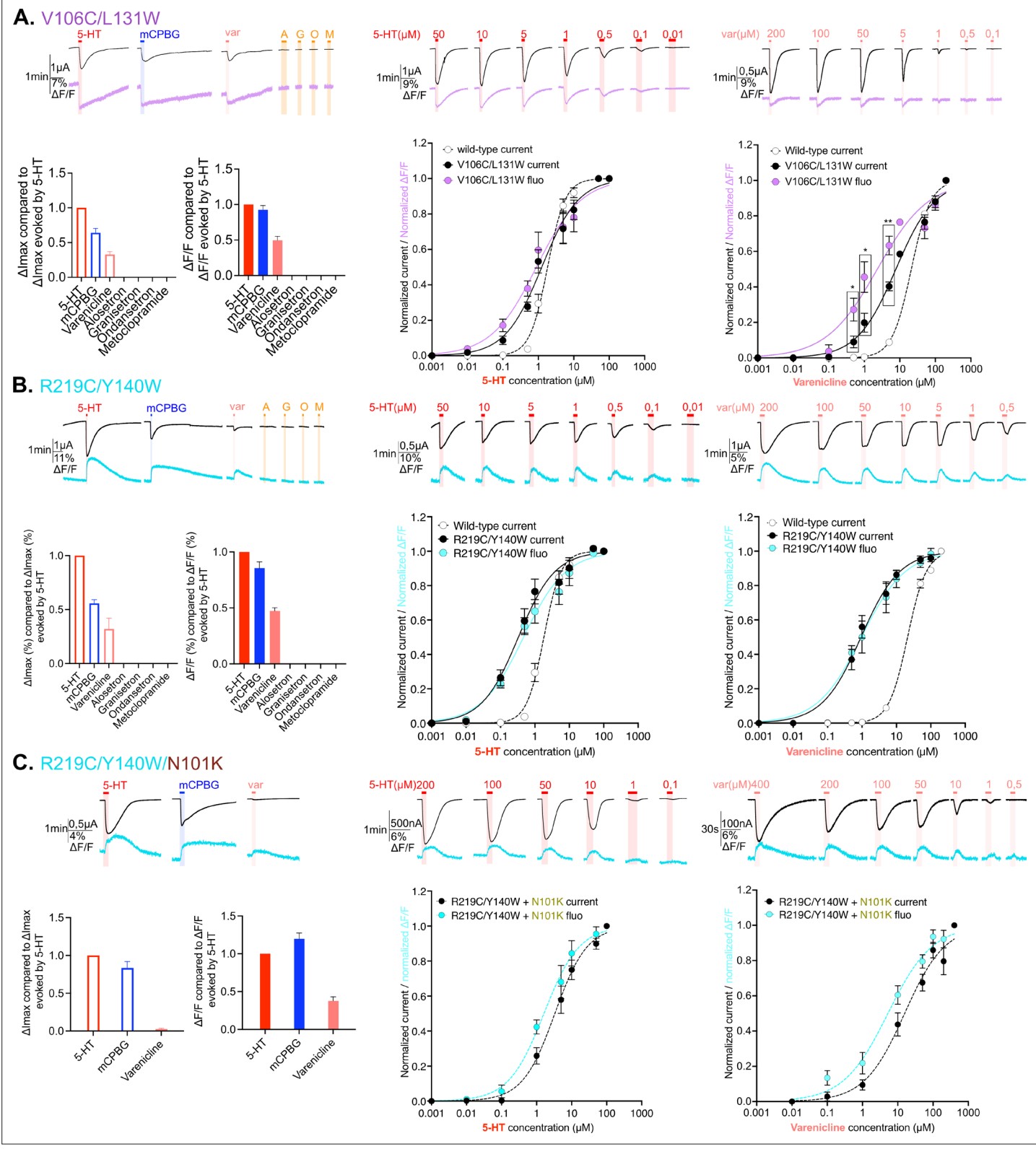

**Figure 3.** Electrophysiological and fluorescence characterization of V106C/L131W, located in the extracellular vestibular area and R219C/Y140W positioned at the interface area between extracellular domain (ECD) and transmembrane domain (TMD). (**A**) Exploration and characterization of the sensor V106C/L131W. Left top panel: macroscopic ligand-gated currents (in black) and fluorescence (in purple) recorded at –60 mV on the construct with the sensor V106C/L131W labeled with MTS-TAMRA evoked by saturating concentrations of agonists (in red, 5-HT; in blue, *m*-chlorophenylbiguanide

*Figure 3 continued on next page*

*Figure 3 continued*

[mCPBG]; in salmon, varenicline) and antagonists (all represented in orange, A: alosetron, G: granisetron, O: ondansetron, M: metoclopramide). Left bottom panel: graphical representation of ligand-induced relative changes of current and fluorescence compared to 5-HT. The normalized values for all the ligands are compared with the mean values obtained for 5-HT. Middle top panel: representative recording of current and fluorescence variations of V106C/L131W labeled with MTS-TAMRA upon different concentrations of perfused 5-HT. Middle bottom panel: dose-response curves for ΔI (black) and ΔF (purple) with mean and SEM (normalized by the maximum current of each oocyte) for application of 5-HT. Right top panel: representative recording of current and fluorescence variations of V106C/L131W labeled with MTS-TAMRA upon different concentrations of perfused varenicline. Right bottom panel: dose-response curves for ΔI (black) and ΔF (magenta) with mean and SEM (normalized by the maximum current of each oocyte) for application of varenicline. Note the significative difference between dose response of current and fluorescence at 0.5, 1, and 5 µM of perfused varenicline (unpaired t-test). (**B**) Exploration and characterization of the sensor R219C/Y140W. Same experiments and legends as for panel A, but the construct here is the sensor R219C/Y140W, current is represented in black and fluorescence in cyan (in trace recordings and dose-response representations). (**C**) Effect of loss-of-function mutation N101K on the sensor R219C/Y140W. Same experiments and legends as for the panel A, but the construct here is the sensor R219C/Y140W with the additional loss-of-function mutation N101K, current is represented in black and fluorescence in cyan (in trace recordings and dose-response representations) (panel A, *Figure 3—source data 1*; panels B and C, *Figure 3—source data 2*). For the entire figure, n are at least 5 and error bars are represented as +/- SEM.

The online version of this article includes the following source data for figure 3:

**Source data 1.** Electrophysiological and fluorescence measurements of the constructs explored in *Figure 3* -1.

**Source data 2.** Electrophysiological and fluorescence measurements of the constructs explored in *Figure 3* -2.

In conclusion, as for orthosteric-located sensors, fluorescent and current signals are well correlated for strong agonists, pointing to a concerted mechanism. In contrast, the partial agonist var on V106C/L131W and especially on R219C/Y140W/N101K shows a decorrelation of fluorescence and current, pointing to the contribution of distinct states endowed with robust ΔF but no ΔI.

## Discussion

In this work, we undertook a screening of 19 mutants of the m5-HT$_{3A}$R with an aim to identify fluorescent sensors within the ECD. Around the orthosteric site, we find that cysteine incorporation often yields ΔF suitable for VCF investigation. In contrast, outside the orthosteric site all single-cysteine mutants investigated failed. This observation parallels a previous VCF screening on the h5-HT$_{3A}$R that identified four suitable sensors around the orthosteric site but none among the seven additional positions tested in the pre-M1 region (*Munro et al., 2019*). Of note, we have previously investigated a sensor on the top of M2 helices on m5-HT$_{3A}$R (*Polovinkin et al., 2018*). In the pLGIC family, the α1-GlyR has also been extensively studied by VCF and many positions were screened for cysteine incorporation (*Shi et al., 2023*; *Pless et al., 2007*; *Pless and Lynch, 2009a*; *Pless and Lynch, 2009b*; *Han et al., 2013*). In contrast to the 5-HT$_3$R, most positions show robust ΔF outside the orthosteric site. This suggests that the α1-GlyR undergoes gating reorganizations of larger amplitude than that of the 5-HT$_3$R. Still, we succeeded in designing two sensors at the vestibule and the ECD-TMD interface by implementing the tryptophan-induced quenching method, further illustrating that this technique is suitable to monitor subtle conformational changes when the fluorophore-quencher pair is properly engineered.

The mapping of the 5-HT$_3$R has enabled us to generate four fluorescence sensors of the protein conformations. They are located at three distant positions within the ECD 3D structure, thereby constituting reference points to infer its global conformation. Simultaneous steady-state concentration-response relationships for the fluorescence and current responses reveal important ligand-elicited protein motions and their relationship with activation.

We first show that agonists and antagonists elicit similar local reorganizations around orthosteric sensors. For both S204C and I160C/Y207W, all tested ligands at saturating concentrations elicit fluorescent dequenching signals. All agonists, partial agonists, granisetron, and ondansetron elicit ΔF of similar amplitude, while alosetron compared to metoclopramide display significantly higher and lower ΔF, respectively. These results are in remarkable concordance with cryo-EM structures. They show that both 5-HT and a series of setron antagonists promote locally a similar reorganization of the orthosteric site, with an inward displacement of the landmark loop C that caps the bound ligand (*Figure 4—figure supplement 1*; *Polovinkin et al., 2018*; *Basak et al., 2018a*). Furthermore, subjecting these structures to molecular dynamic simulations suggests that different antagonists promote different

**Table 1.** EC$_{50}$ values for current (EC$_{50}$c) and fluorescence (EC$_{50}$f) responses to agonists (5-HT, *m*-chlorophenylbiguanide [mCPBG], varenicline) at labeled and unlabeled m5-HT$_{3A}$ mutants.

The top part of the table represents the characterization of the sensors with different ligands and the associated controls (rows 1–13) and the second part the additional allosteric mutations added (rows 14–16).

| Construct | Molecule | EC$_{50}$c (μM) ± SEM | n$_{Hill}$c ± SEM | EC$_{50}$f (μM) ± SEM | n$_{Hill}$f ± SEM | n |
|---|---|---|---|---|---|---|
|  | 5-HT | 1.83±0.12 | 1.74±0.13 | / | / | 8 |
|  | mCPBG | 1.13±0.15 | 1.02±0.14 | / | / | 6 |
| Wild-type (WT) | Varenicline | 21.09±1.03 | 1.595±0.07 | / | / | 5 |
| WT+MTS-TAMRA | 5-HT | 3.43±0.24 | 1.98±0.12 | / | / | 5 |
|  | 5-HT | 31.22±2.76 | 1.69±0.18 | 10.73±0.62 | 1.56±0.14 | 7 |
|  | mCPBG | 7.99±0.58 | 1.19±0.105 | 7.46±0.755 | 0.92±0.08 | 8 |
| I160C/Y207W+MTS-TAMRA | Varenicline | No current | / | 16.80±0.76 | 1.29±0.06 | 7 |
|  | 5-HT | 47.33±4.72 | 1.19±0.135 | / | / | 6 |
|  | mCPBG | 2.73±0.17 | 1.59±0.11 | / | / | 8 |
| I160C/Y207W unlabeled | Varenicline | No current | / | / | / | / |
| I160C+MTS-TAMRA | 5-HT | 31.01±1.64 | 1.73±0.11 | / | / | 5 |
|  | 5-HT | 1.78±0.13 | 1.39±0.11 | 2.86±0.248 | 1.51±0.14 | 6 |
|  | mCPBG | 1.45±0.18 | 0.94±0.10 | 2.90±0.40 | 0.80±0.08 | 6 |
| S204C+MTS-TAMRA | Varenicline | 20.67±0.705 | 1.48±0.05 | 16.31±0.78 | 1.29±0.06 | 7 |
|  | 5-HT | 1.93±0.25 | 1.08±0.12 | / | / | 6 |
|  | mCPBG | 0.54±0.02 | 1.94±0.16 | / | / | 6 |
| S204C unlabeled | Varenicline | 17.105±1.12 | 1.385±0.07 | / | / | 7 |
|  | 5-HT | 1.25±0.155 | 0.84±0.08 | 0.88±0.16 | 0.68±0.08 | 5 |
| V106C/L131W+MTS-TAMRA | Varenicline | 7.745±0.775 | 0.80±0.05 | 2.33±0.45 | 0.60±0.06 | 5 |
|  | 5-HT | 1.91±017 | 0.99±0.07 | / | / | 9 |
| V106C/L131W unlabeled | Varenicline | 6.06±0.32 | 1.47±0.12 | / | / | 5 |
| V106C+MTS-TAMRA | 5-HT | 4.47±0.55 | 1.08±0.13 | / | / | 5 |
|  | 5-HT | 0.33±0.05 | 0.82±0.10 | 0.47±0.07 | 0.71±0.07 | 8 |
| R219C/Y140W+MTS-TAMRA | Varenicline | 0.94±0.12 | 0.77±0.08 | 0.96±0.19 | 0.68±0.10 | 6 |
|  | 5-HT | 0.60±0.01 | 0.94±0.10 | / | / | 7 |
| R219C/Y140W unlabeled | Varenicline | 1.575±0.28 | 0.875±0.12 | / | / | 6 |
| R219C+ MTS-TAMRA | 5-HT | 0.11±0.03 | 0.92±0.21 | 0.185±0.04 | 0.94±0.15 | 6 |
| WT+N101K | 5-HT | 66.67±3.11 | 1.95±0.185 | / | / | 6 |
|  | 5-HT | 57.11±1.49 | 2.33±0.17 | 12.19±0.90 | 1.285±0.11 | 6 |
|  | Varenicline | No current | / | 25.30±2.46 | 0.94±0.07 | 7 |
| S204C+N101K | mCPBG | 13.10±-0.79 | 1.345±0.10 | 4.365±0.495 | 0.72±0.05 | 7 |
|  | 5-HT | 3.32±0.39 | 0.94±0.10 | 1.62±0.27 | 0.86±0.11 | 9 |
| R219C/Y140W+N101K | Varenicline | 15.65±2.70 | 0.76±0.09 | 5.25±0.95 | 0.69±0.07 | 6 |

**Table 2.** Current and fluorescence maximum evoked by ligands (agonist and antagonists) on m5-HT$_{3A}$ mutants.

| Construct | Molecule | $\Delta Imax_{molecule}$ compared to $\Delta Imax_{5-HT}$ ± SEM | $\Delta Fmax_{molecule}$ compared to $\Delta Fmax_{5-HT}$ ± SEM | n |
|---|---|---|---|---|
| | mCPBG | 0.92±0.03 | / | |
| | Varenicline | 0.39±0.06 | / | |
| | Alosetron | 0 | / | |
| | Granisetron | 0 | / | |
| | Ondasetron | 0 | / | |
| Wild-type | Metoclopramide | 0 | / | 6 |
| | mCPBG | 0.89±0.04 | 1.09±0.105 | |
| | Varenicline | 0.33±0.06 | 1.12±0.16 | |
| | Alosetron | 0 | 1.44±0.20 | |
| | Granisetron | 0 | 1.14±0.19 | |
| | Ondasetron | 0 | 0.78±0.13 | |
| S204C+MTS-TAMRA | Metoclopramide | 0 | 0.52±0.08 | 6 |
| | mCPBG | *5-HT – 0.02±0.004 | 1.21±0.06 | |
| | Varenicline | *0 | 1.52±0.06 | |
| | Alosetron | *0 | 2.26±0.10 | |
| | Granisetron | *0 | 1.12±0.07 | |
| | Ondasetron | *0 | 1.27±0.03 | |
| I160C/Y207W+MTS-TAMRA | Metoclopramide | *0 | 0.60±0.04 | 8 |
| | mCPBG | 0.64±0.06 | 0.93±-0.06 | |
| | Varenicline | 0.33±0.04 | 0.49±0.06 | |
| | Alosetron | 0 | 0 | |
| | Granisetron | 0 | 0 | |
| | Ondasetron | 0 | 0 | |
| V106C/L131W+MTS-TAMRA | Metoclopramide | 0 | 0 | 6 |
| | mCPBG | 0.62±0.07 | 0.94±0.09 | |
| | Varenicline | 0.34±0.08 | 0.47±0.02 | |
| | Alosetron | 0 | 0 | |
| | Granisetron | 0 | 0 | |
| | Ondasetron | 0 | 0 | |
| R219C/Y140W+MTS-TAMRA | Metoclopramide | 0 | 0 | 6 |
| | mCPBG | 1.80±0.11 | 1.23±0.05 | |
| S204C+N101K-MTS-TAMRA | Varenicline | 0 | 1.73±0.05 | 8 |
| | mCPBG | 0.83±0.08 | 1.20±0.08 | |
| R219C/Y140W+N101K-MTS-TAMRA | Varenicline | 0.03±0.01 | 0.38±0.06 | 7 |

Here : *comparison with values obtained with mCPBG (in contrast with all the other values in this table that where compared to values obtained with 5-HT).

/ : not applicable.

degrees of local conformational changes (**Basak et al., 2019**; **Basak et al., 2020**). Among setrons, alosetron is the one promoting the largest conformational effects, consistent with VCF data. Of note, the ligand-elicited dequenching signal of the I160C/Y207W is also consistent with the cryo-EM structures, since the capping of loop C is predicted to reduce the accessibility of the indole side chain to the rhodamine dye grafted on the other side of the loop. Interestingly, three other fluorescence sensors around the orthosteric site were already reported on the human 5-HT$_{3A}$R on loop C (M223C), D (Y89C), and E (Q146C). They also report that agonists and antagonists equally elicit fluorescence dequenching, with ΔF intensities strongly ligand-dependent for loops C and E, but not for loop D (**Munro et al., 2019**).

We also show that antagonist-elicited reorganizations do not spread to the vestibular and ECD-TMD sensors. Indeed, the sensors at the ECD-TMD interface (R219C/Y140W) and at the middle vestibule (V106C/L131W) show robust ΔF upon perfusion of agonists, but no effect for antagonists. Cryo-EM structures show no setron-elicited change at the ECD-TMD interface, especially no motion of the M2-M3 loop that lies in between the attachment points of the fluorophore and the indole of the tryptophan in R219C/Y140W (**Figure 4—figure supplement 1**; **Polovinkin et al., 2018**; **Basak et al., 2020**; **Zarkadas et al., 2020**). Likewise, setrons elicit small reorganizations of the architecture of the vestibule (measured as a slightly increased radius of the constriction at K108 and D105), while 5-HT elicits stronger reorganizations at this level. Overall, VCF data are in excellent agreement with cryo-EM data, providing evidence that high-resolution structures of setron-inhibited states represent pertinent snapshots of actual allosteric states at the plasma membrane.

Steady-state concentration-response relationships support that strong agonists elicit an apparently concerted reorganization of the global protein structure. Indeed, for most constructs, those agonists eliciting the maximal response currents among all tested agonists show no difference in ΔF and ΔI concentration-response curves. This is the case for 5-HT, that maximally activates the S204C, V106C/L131W, and R219C/Y140W sensors and for mCPBG that maximally activates the S204C and I160C/Y207W sensors. This suggests a concerted mechanism where local motions around the four sensors happen simultaneously with the pore opening. These data are thus accounted by a simple two-state model, where the receptor is in equilibrium between a resting and an active state. VCF measurements are also remarkably coherent with the 5-HT-bound structures showing an open pore: PDB 6HIN (called F), PDB 6DG8 (called state 2), and PDB Y5A (called serotonin-bound state). Indeed, all of them show symmetrical and global reorganizations of the ECD involving loop C capping, vestibular reorganization, and M2-M3 reorganization (**Figure 4—figure supplement 1**; **Polovinkin et al., 2018**; **Basak et al., 2018a**; **Zhang et al., 2021**).

However, we found that partial agonists stabilize additional intermediate conformations characterized by fluorescence variations but no current, especially when loss-of-function mutations are engineered. Intermediate conformations are observed for the two sensors outside the orthosteric site. Indeed, for R219C/Y140W sensor in combination with N101K, var elicits at saturation only 3% of 5-HT currents but promotes a robust ΔF signal corresponding to 38% of that of 5-HT. In addition, the ΔF concentration-response curve appears slightly left-shifted as compared to the ΔI curve, indicating that the ΔF/ΔI ratio is even higher at low var concentrations. These data clearly show that var stabilizes in majority intermediate conformations where the ECD-TMD interface has moved, generating a ΔF, but where the channel remains closed. A similar although milder 'intermediate' phenotype is seen for var on the V106C/L131W sensor.

Within the gating pathway, these intermediate conformations may a priori contribute either to the activation transition (pre-active state) or to the desensitization transition (fast or slow desensitized states). However, VCF data strongly argue for the former hypothesis. Indeed, for all experiments including var partial activation of V106C/L131W, the rise times of ΔF are in the same range to that of ΔI, desensitization being much slower and associated with no significant ΔF. This shows that the intermediate states are recruited during or concomitant with the activation phase. This is in line with the general observation that partial agonists currents are strongly increased by positive allosteric modulators, indicating that the electrophysiologically silent conformations they promote are activatable and therefore not desensitized (**Felt et al., 2024**).

Two 5-HT-bound structures of m5-HT$_{3A}$ displaying a closed channel sustain the intermediate reorganizations seen by VCF. Indeed, conformations called I1 (PDB 6HIO) and state 1 (PDB 6DG7) feature rearrangements at the ECD and interface between ECD/TMD similar to that of conformation displaying

an open channel, but with non-conducting pore (*Figure 4—figure supplement 1*; *Polovinkin et al., 2018*; *Basak et al., 2018a*). It is noteworthy that these states were initially proposed to represent either pre-active or desensitized states, and that these two possibilities could not be distinguished without ambiguity. However, indirect experiments of substituted cysteine accessibility method (SCAM) suggested that desensitization involves weak reorganizations of the upper part of the channel that holds the activation gate, arguing for the pre-active state hypothesis (*Polovinkin et al., 2018*). Combined SCAM and VCF data thus provide compelling evidence for annotation as pre-active state. Interestingly, during the revision of this article, several cryo-EM structures of m5-HT$_{3A}$ in complex with partial agonists SMP-100 and ALB-148471 were published. In full agreement with our VCF data, partial agonists are found to stabilize the protein in both active and pre-active-like conformation (*Felt et al., 2024*).

Intermediate conformations are also observed on many occasions for the sensors near the orthosteric site. This is the case with the var partial agonist on S204C, which produces a lower maximum current variation (ΔImax) than that caused by 5-HT but a higher ΔFmax. The phenotype is exacerbated in the presence of N101K, 5-HT, and mCPBG showing intermediate conformations at sub-saturating concentrations, while var promote no current while still evoking robust ΔF. Of note, on this mutant, mCPBG is the most efficient agonist, but it remains possible that it has a partial agonistic character due to the strong loss-of-function N101K mutation. In this context, the sensor I160C/Y207W shows a loss-of-function phenotype resembling that of S204C/N101K with comparable relative position of the ΔI and ΔF dose-response curves. Correlating the motions at the orthosteric site with known cryo-EM structures is, however, difficult. Indeed, two hypotheses can be sustained by these results: the intermediate conformations revealed by these sensors can correspond either to those stabilized by setrons or to the pre-active conformations discussed above.

Altogether, VCF data highlight a progressive propagation of the signal following ligand binding. Setrons elicit local reorganizations shown by the sensors located around the orthosteric site, partial agonists elicit local reorganizations at the four sensors indicating a motion of the whole ECD with partial pore opening, and strong agonists elicit reorganizations detected by all four sensors together with channel opening. VCF data thus identify four families of conformations endowed with distinct ΔI/ΔF signatures contributing to signal transduction: resting-like apo, setron-inhibited, partial agonist-elicited pre-active and active states. The topological information given by the various sensors are remarkably consistent with the gallery of high-resolution structures solved thus far. Indeed, the data from fluorescence partners are consistent with respectively apo, setron-bound, 5-HT-bound-closed, and 5-HT-bound-open conformations. This provides important information allowing reasonable functional annotation of the various structures to physiological states in a membrane environment, at least regarding the ECD. *Figure 4* shows a speculative four-state model as a framework integrating the whole set of data. In addition, VCF data give insights in the action of partial agonists, that do not exclusively stabilize the active state and document the phenotypes of various allosteric mutations.

A key observation of the study is the identification of pre-active intermediates that are favored upon binding of partial agonists and/or in the presence of loss-of-function mutations. It is noteworthy that single-channel kinetic analyses of pLGICs early showed that the activation transition pathway involves multiple intermediated states. Analysis of numerous mutants of the muscle nAChR analyzed by REFERs (rate equilibrium linear free energy relationships) suggested a multistep reorganization that initiates in the orthosteric site and progressively spreads to the ion channel gate (*Grosman et al., 2000*). Analysis on GlyRs and nAChRs detected late intermediate states called 'flip' (*Burzomato et al., 2004*) or 'primed' (*Mukhtasimova et al., 2009*), that are favored by loss-of-function mutations (*Plested et al., 2007*; *Lape et al., 2012*), while a non-conducting intermediate state has been proposed from kinetic models of a high-conductance mutant of 5-HT$_3$R (*Corradi et al., 2009*).

More recently, fluorescence and VCF studies identified intermediate conformations for nAChRs, α1-GlyRs, and the bacterial homolog GLIC (*Shi et al., 2023*; *Dahan et al., 2004*; *Mourot et al., 2008*; *Menny et al., 2017*; *Lefebvre et al., 2021*). Of note, pre-active intermediates were unraveled by a fluorescence quenching pair at strictly homologous positions at the ECD-TMD interface of 5-HT$_3$R and α1-GlyRs. Indeed, residues R219C/Y140W are homologous to another sensor pair Q219C/K143W engineered on the α1-GlyR. MTS-TAMRA-labeled Q219C/K143W also reports a non-conducting intermediate in the pathway toward activation, with a fluorescence variation revealing an early reorganization of the ECD-TMD interface (*Shi et al., 2023*). Molecular dynamic simulations starting from a

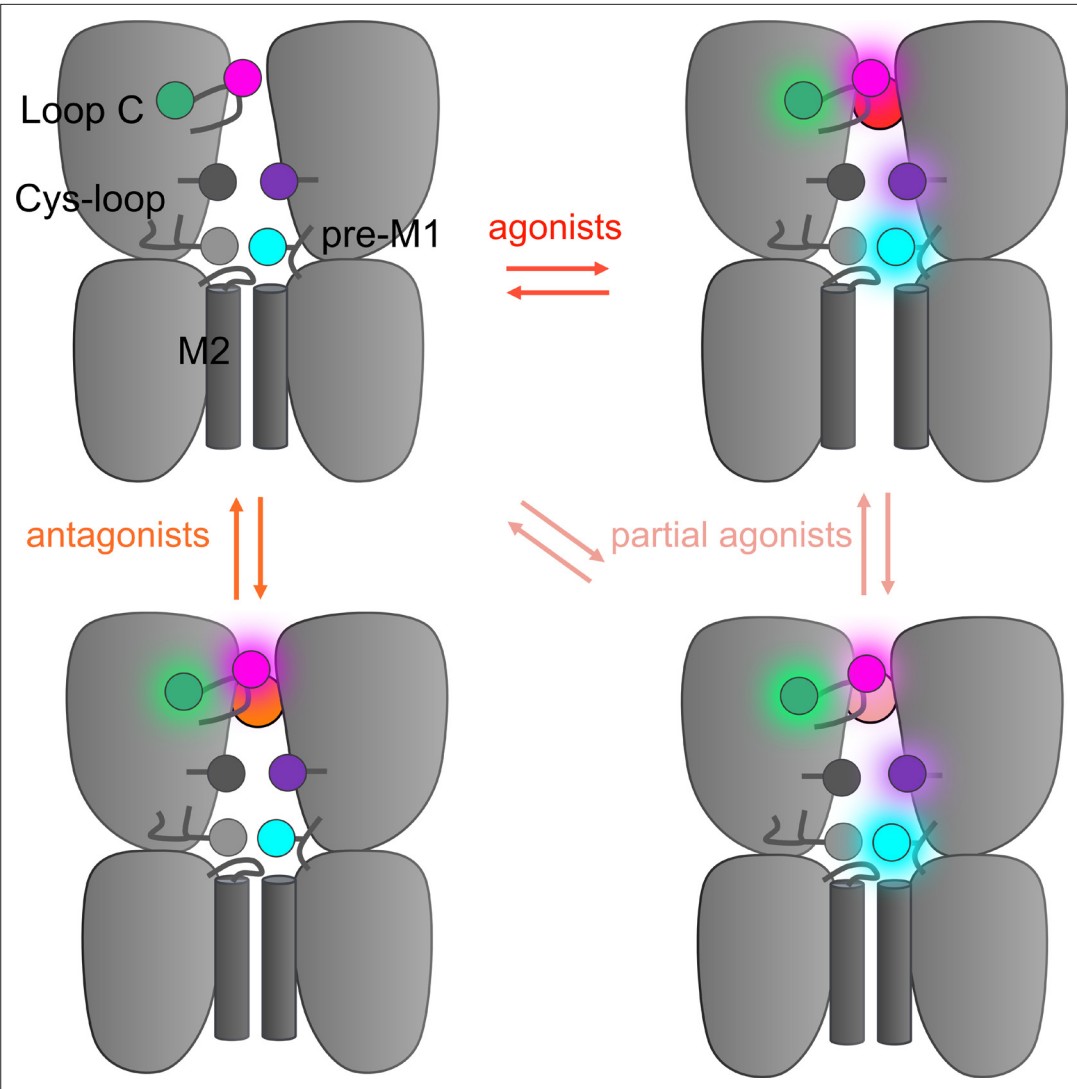

**Figure 4.** Hypothetical integrative model of voltage-clamp fluorometry (VCF) data. Schematic representation of the 5-HT₃R representing two subunits in side-view with the orthosteric site and ion channel M2 helices highlighted. The four fluorescent sensors are represented as hexagons following the color code of *Figure 1* (S204C in magenta, I160C/Y207W in green, V106C/L131W in purple, R219C/Y140W in cyan). Ligand-elicited fluorescence changes are represented as a light halo. VCF data identifies four different conformations whose fluorescence patterns match known high-resolution structures. These conformations are called resting (matching apo structures, PDB 4PIR, 6BE1, 6H5B, 6Y59), inhibited (matching setron-bound structures, PDB 6HIS, 6W1J, 6W1M, 6W1Y, 6Y1Z), intermediate (matching I1 and state 1, PDB 6HIO, 6DG7), and active (matching F, state 2, and open, PDB 6HIN, 6DG8, 6Y5A).

The online version of this article includes the following figure supplement(s) for figure 4:

**Figure supplement 1.** Structural comparison of representative cryo-electron microscopy (cryo-EM) structures.

taurine-bound-closed cryo-EM structure was found to recapitulate the pharmacological properties observed by VCF, suggesting that the intermediate conformations correspond to a highly dynamic family of conformations featuring an active-like ECD but a resting-like pore (*Yu et al., 2021*). Altogether, VCF shows that virtually all pLGICs appear to share a common global mechanism of gating where the proteins visit a family of structurally dynamic intermediates showing an active-like ECD and a resting-like TMD conformation. The present work thus extends this idea to the 5-HT₃AR, together with providing structural blueprints for cryo-EM structural annotation.

In conclusion, by monitoring simultaneously electrophysiologically silent and active conformations of the 5-HT₃R, VCF allowed to characterize the mechanisms of action of allosteric effectors

and allosteric mutations. We found that the strength of the agonist correlated to a degree with propagation of this fluorescence change beyond the local site of neurotransmitter binding, unraveling intermediate conformations allowing proposing a structure-based functional annotation of known high-resolution structures. Besides validating the mechanism underlying inhibition by setrons, these data unravel a unique mode of action of partial agonists to promote distinct intermediate conformations. The various fluorescence sensors developed will be valuable in future mutational analysis and characterization of drugs acting at the 5-HT$_3$R that hold promise for clinical use.

# Materials and methods

**Key resources table**

| Reagent type (species) or resource | Designation | Source or reference | Identifiers | Additional information |
|---|---|---|---|---|
| Gene (*Mus musculus*) | *HTR3A*, mouse 5-HT3A | Derived from doi: https://doi.org/10.1038/s41586-018-0672-3 | Uniprot: P23979 | |
| Biological sample (*Xenopus laevis*) | *Xenopus laevis* dissociated oocytes | Ecocyte Bioscience (Dortmund-Germany) | | |
| Biological sample (*Xenopus laevis*) | *Xenopus laevis* ovarian fragments | Portsmouth European *Xenopus* Resource Centre | | |
| Recombinant DNA reagent | Mouse 5-HT$_{3A}$ in pcDNA5 vector | DOI: https://doi.org/10.1038/s41586-018-0672-3 | | |
| Chemical compound, drug | Serotonin (5-HT), 5-hydroxytryptamin hydrochloride | Merck (Sigma) | CAS: 153-98-0 | |
| Chemical compound, drug | mCPBG hydrochloride | Merck (Sigma) | CAS: 2113-05-5 | |
| Chemical compound, drug | Varenicline tartrate (var) | Merck (Sigma) | CAS: 375815-87-5 | |
| Chemical compound, drug | Alosetron hydrochloride | Merck (Sigma) | CAS: 122852-69-1 | |
| Chemical compound, drug | Granisetron hydrochloride | Merck (Sigma) | CAS: 107007-99-8 | |
| Chemical compound, drug | Ondansetron hydrochloride dihydrate | Merck (Sigma) | CAS: 103639-04-9 | |
| Chemical compound, drug | Metoclopramide | Merck (Sigma) | CAS: 364-62-5 | |
| Chemical compound, drug | MTS-TAMRA | Clinisciences | Catalog Number: 91030 | MTS-5(6)-carboxytetramethylrhodamine, mixed isomers |
| Chemical compound, drug | DMSO | Merck (Sigma) | CAS: 67-68-5 | Anhydrous |
| Software, algorithm | Prism | GraphPad | | |
| Software, algorithm | Clampfit | Molecular Devices | | |
| Other | Custom-made recording chamber for VCF | DOI: https://doi.org/10.1038/s41467-023-36471-7 | | Custom compartment chamber for VCF recordings used in this study |

## Materials

The ligands used for perfusion: 5-hydroxytryptamine hydrochloride (5-HT, serotonin), mCPBG, var tartrate, alosetron, granisetron, ondansetron, and metoclopramide have been purchased from Merck (Sigma). Hydrosoluble ligands were directly solubilized in distilled water or in recording solution, aliquoted at high concentrations, kept in −20°C and used freshly for dilution on the day of experiment. Less soluble ligands (ondansetron) were first dissolved in DMSO, aliquoted and kept at −20°C, and then diluted in recording solutions at final concentrations not exceeding 1% of final DMSO on the

day of experiment. The fluorescent dye MTS-TAMRA was purchased from Clinisciences, aliquoted in DMSO, kept at –20°C, and freshly used for labeling.

## Site-directed mutagenesis

All mutations were carried into the gene of the mouse 5-HT$_{3A}$ (described in *Polovinkin et al., 2018*, where strep-tags were removed), kindly provided by Hugues Nury (Institut de Biologie Structurale, Grenoble, France). The mutations were introduced by site-directed mutagenesis via PCR (CloneAmp Hifi, Takara). All mutations were assessed by complete sequencing of the gene (Eurofins Genomics). We have chosen to work with the mouse receptor because most of the available atomic structure have been obtained for m5-HT$_{3A}$ receptors.

## Oocytes handling

*X. laevis* ovarian fragments (TEFOR PARIS SACLAY CNRS UAR2010/INRAE UMS1451) and dissociated stage VI oocytes (Ecocyte Biosciences, Germany) were used in the context of this study. Concerning ovarian fragments, oocytes were dissociated following enzymatic treatment with collagenase II (1 mg/mL; 1 hr at room temperature in gentle agitation) (Thermo Fisher) in ORII solution (in mM: 82.5 NaCl, 2.5 KCl, 1 MgCl$_2$, 5 HEPES, pH adjusted to 7.6 with NaOH). Selected oocytes were then handled in Barth's solution (in mM: 88 NaCl, 1 KCl, 0.33 Ca(NO$_3$)$_2$, 0.41 CaCl$_2$, 0.82 MgSO$_4$, 2.4 NaHCO$_3$, 10 HEPES, pH adjusted to 7.6 with NaOH) at 18°C.

## cDNA injection

cDNA encoding for m5-HT$_{3A}$ constructs were injected into the nucleus of oocytes (100 ng) with cDNA encoding for eGFP as a reporter of correct injection (25 ng) by an air injection system (Nanoject II, Drummond). Oocytes were then incubated at 18°C and used to be recorded 48–96 hr after injection.

## Labeling of mutants

Oocytes were incubated for 20 min at room temperature with a solution containing 10 µM of MTS-TAMRA (in DMSO) and 1–10 µM of 5-HT (in ND96 0% calcium) allowing the final concentration of DMSO to not exceed 0.1%. The oocytes were then rinsed three times with perfusion solution and recorded during the following 2 hr.

## Voltage-clamp fluorometry

Oocytes were placed in a custom-made recording chamber (described in *Shi et al., 2023*). It allows to record and perfuse the same part of the animal pole that faces the perfusion and the inverted microscope. Oocytes were continuously perfused with freshly made ND96 0% calcium (in mM: 96 NaCl, 2 KCl, 5 HEPES, 1 MgCl$_2$, 1 HEPES, 1 EGTA, and pH was adjusted at 7.6 with NaOH). To perform recordings, microelectrodes (borosilicate glass with filament BF150-110-7.5, WPI) of resistances comprised between 0.2 and 2 mΩ (pipette puller PC-10, Narishige) were used and oocytes were clamped at –60 mV for all the experiments. Recordings are performed with a GeneClamp 500 voltage patch-clamp amplifier (Axon Instruments) and a 1400 A digitizer (Axon Instruments) with Clampex 10.6 software (Molecular Devices). Recorded currents were sampled at 2 kHz and filtered at 500 Hz. The fluorescence emission was recorded via a FF01-543_22 bandpass filter (Semrock) and collected by a photo-multiplicator (H10722, Hamamatsu). The intensity of irradiation of the LEDs (pE-4000 CoolLED) and of the sensitivity of detection of the PMT were kept at the same level for all the experiments. For the screening experiment, 5-HT at 50 and 100 µM were perfused (around 10 s). Dose-response curves of 5-HT, mCPBG, and var were adapted to the phenotype of the construct (gain- or loss-of-function) and used at concentrations stated in legends of graphs from the figures. Each concentration has been repeated twice for establishment of the dose-response curves. For the pharmacological experiments, used compounds were perfused on the same oocyte at saturating concentration (5-HT and mCPBG at 200 µM, var at 400 µM, antagonists used have low nanomolar affinities and used here all at 3 µM concentration).

## Two-electrode voltage clamp

Currents from impaled oocytes expressing m5-HT$_{3A}$ constructs were obtained under ND96 0% calcium perfusion. Currents were recorded by a Warner OC-725C amplifier and digitized by a Digidata 1550A

with Clampex 10 software (Molecular Devices). Currents were sampled at 500 Hz and filtered at 100 Hz. The voltage clamp is maintained at –60 mV during all experiments. The TEVC setup has a faster perfusion than the VCF setup, as described in a previous paper from our lab (*Gielen et al., 2020*). To characterize some of the desensitization properties, long application of 5-HT (45 s) at several concentrations (10, 50, 100, 300, and 500 µM) have been applied to calculate the remaining currents after 45 s application and compared to WT.

## Analysis of results

Current and fluorescence analyses were made with Clampfit (Molecular Devices, Sunnyvale, CA, USA). Dose-response curves, $EC_{50}$, and Hill coefficients are obtained by the normalization of serotonin-induced currents to the maximal current followed by the fitting of the data by one-site Hill equation (GraphPad Prism). Error bars on figures represents ± SEM. Results were obtained from at minima five different oocytes from at minima two different batches of oocytes. Rise time constants and desensitization constants have been calculated in Clampfit by mono-exponential fittings of signals. Statistical analyses have been made with Prism (unpaired t-tests).

## Acknowledgements

The work was supported by the ERC (Grant no. 788974, Dynacotine). The authors would like to thank Hugues Nury for the kind gift of mouse 5-HT$_3$ cDNA, Kate Dunning, Solène Lefebvre, and Thomas Grutter for the critical reading of the manuscript.

## Additional information

### Funding

| Funder | Grant reference number | Author |
|---|---|---|
| European Research Council | 788974 | Pierre-Jean Corringer Laurie Peverini Sophie Shi |

The funders had no role in study design, data collection and interpretation, or the decision to submit the work for publication.

### Author contributions

Laurie Peverini, Conceptualization, Data curation, Formal analysis, Validation, Investigation, Visualization, Methodology, Writing - original draft, Writing - review and editing; Sophie Shi, Karima Medjebeur, Validation, Investigation; Pierre-Jean Corringer, Conceptualization, Supervision, Funding acquisition, Investigation, Writing - original draft, Project administration, Writing - review and editing

### Author ORCIDs

Laurie Peverini (ID) http://orcid.org/0000-0002-7992-1245
Pierre-Jean Corringer (ID) http://orcid.org/0000-0002-4770-430X

Reviewer #1 (Public Review): https://doi.org/10.7554/eLife.93174.3.sa1
Reviewer #2 (Public Review): https://doi.org/10.7554/eLife.93174.3.sa2
Reviewer #3 (Public Review): https://doi.org/10.7554/eLife.93174.3.sa3
Author response https://doi.org/10.7554/eLife.93174.3.sa4

## Additional files

### Supplementary files

• Supplementary file 1. Contains Table S1. Values of $\tau_{rise}$ obtained from rise time kinetic analyses for current and fluorescence on labeled m5-HT$_{3A}$ mutants.

• MDAR checklist

## Data availability

All the data generated and analyzed during this study have been included in the manuscript (*Tables 1 and 2*) and supporting files.

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
