## [Editor Report · eLife assessment]

This **valuable** study applies voltage clamp fluorometry to provide new information about the function of serotonin-gated ion channels 5-HT3AR. The authors **convincingly** investigate structural changes inside and outside the orthosteric site elicited by agonists, partial agonists, and antagonists, helping to annotate existing cryo-EM structures. This work confirms that the activation of 5-HT3 receptors is similar to other members of this well-studied receptor superfamily. The work will be of interest to scientists working on channel biophysics but also drug development targeting ligand-gated ion channels.

---

## [Referee Report · Reviewer #1 (Public Review)]

Summary:

This study brings new information about the function of serotonin-gated ion channels 5-HT3AR, by describing the conformational changes undergoing during ligands binding. These results can be potentially extrapolated to other members of the Cys-loop ligand-gated ion channels. By combining fluorescence microscopy with electrophysiological recordings, the authors investigate structural changes inside and outside the orthosteric site elicited by agonists, partial agonists, and antagonists. The results are convincing and correlate well with the observations from cryo-EM structures. The work will be of important significance and broad interest to scientists working on channel biophysics but also drug development targeting ligand-gated ion channels.

Strengths:

The authors present an elegant and well-designed study to investigate the conformational changes on 5-HT3AR where they combine electrophysiological and fluorometry recordings. They determined four positions suitable to act as sensors for the conformational changes of the receptor: two inside and two outside the agonist binding site. They make a strong point showing how antagonists produce conformational changes inside the orthosteric site similarly as agonists do but they failed to spread to the lower part of the ECD, in agreement with previous studies and Cryo-EM structures. They also show how some loss-of-function mutant receptors elicit conformational changes (changes in fluorescence) after partial agonist binding but failed to produce measurable ionic currents, pointing to intermediate states that are stabilized in these conditions. The four fluorescence sensors developed in this study may be good tools for further studies on characterizing drugs targeting the 5-HT3R. The major conclusions of the manuscript seem well justified.

Weaknesses:

Weaknesses have been very well addressed during the review process.

---

## [Referee Report · Reviewer #2 (Public Review)]

Summary:

This study focuses on the 5-HT3 serotonin receptor, a pentameric ligand-gated ion channel important in chemical neurotransmission. There are many cryo-EM structures of this receptor with diverse ligands bound, however assignment of functional states to the structures remains incomplete. The team applies voltage-clamp fluorometry to measure, at once, both changes in ion channel activity, and changes in fluorescence. Four cysteine mutants were selected for fluorophore labeling, two near the neurotransmitter site, one in the ECD vestibule, and one at the ECD-TMD junction. Agonists, partial agonists, and antagonists were all found to yield similar changes in fluorescence, a proxy for conformational change, near the neurotransmitter site. The strength of the agonist correlated to a degree with propagation of this fluorescence change beyond the local site of neurotransmitter binding. Antagonists failed to elicit a change in fluorescence in the vestibular of the ECD-TMD junction sites. The VCF results further turned up evidence supporting intermediate (likely pre-active) states.

Strengths:

The experiments appear rigorous, the problem the team tackles is timely and important, the writing and the figures are for the most part very clear. We sorely need approaches orthogonal to structural biology to annotate conformational states and observe conformational transitions in real membranes- this approach, and this study, get right to the heart of what is missing.

Weaknesses:

The weaknesses in the study itself are overall minor, I only suggest improvements geared toward clarity. What we are still missing is application of an approach like this to annotate the conformation of the part of the receptor buried in the membrane; there is an important debate about which structure represents which state, and that is not addressed in the current study.

---

## [Referee Report · Reviewer #3 (Public Review)]

Summary:

The authors have examined the 5-HT3 receptor using voltage clamp fluorometry, which enables them to detect structural changes at the same time as the state of receptor activation. These are ensemble measurements, but they enable an impressive scheme of the action of different agonists and antagonists to be built up. The growing array of structural snapshots of 5-HT3 receptors is used to good effect to understand the results.

Strengths:

The combination of rigorously tested fluorescence reporters with oocyte electrophysiology across a large panel of ligands is a solid development for this receptor type.

Weaknesses:

In their revision, the authors corrected all the weaknesses of the original submission.

---

## [Author Response]

The following is the authors’ response to the original reviews.

**eLife assessment**
This valuable study applies voltage clamp fluorometry to provide new information about the function of serotonin-gated ion channels 5-HT3AR. The authors convincingly investigate structural changes inside and outside the orthosteric site elicited by agonists, partial agonists, and antagonists, helping to annotate existing cryo-EM structures. This work confirms that the activation of 5-HT3 receptors is similar to other members of this well-studied receptor superfamily. The work will be of interest to scientists working on channel biophysics but also drug development targeting ligand-gated ion channels.
**Public Reviews:**
All reviewers agreed that these results are solid and interesting. However, reviewers also raised several concerns about the interpretation of the data and some other aspects related to data analysis and discussion that should be addressed by the authors. Essential revisions should include:(1) Please try to explicitly distinguish between a closed pore and a resting or desensitized state of the pore, to help in clarity.(2) Add quantification of VCF data (e.g. sensor current kinetics, as suggested by reviewer #2) or better clarify/discuss the VCF quantitative aspects that are taken into account to reach some conclusions (reviewer #3).(3) Review and add relevant foundational work relevant to this study that is not adequately cited.(4) Revise the text according to all recommendations raised by the reviewers and listed in the individual reviews below.

We have revised the text to address all four points. See the answers to referees’ recommendations.

**Reviewer #1 (Public Review):**
Summary:This study brings new information about the function of serotonin-gated ion channels 5-HT3AR, by describing the conformational changes undergoing during ligands binding. These results can be potentially extrapolated to other members of the Cys-loop ligand-gated ion channels. By combining fluorescence microscopy with electrophysiological recordings, the authors investigate structural changes inside and outside the orthosteric site elicited by agonists, partial agonists, and antagonists. The results are convincing and correlate well with the observations from cryo-EM structures. The work will be of important significance and broad interest to scientists working on channel biophysics but also drug development targeting ligand-gated ion channels.Strengths:The authors present an elegant and well-designed study to investigate the conformational changes on 5-HT3AR where they combine electrophysiological and fluorometry recordings. They determined four positions suitable to act as sensors for the conformational changes of the receptor: two inside and two outside the agonist binding site. They make a strong point showing how antagonists produce conformational changes inside the orthosteric site similarly as agonists do but they failed to spread to the lower part of the ECD, in agreement with previous studies and Cryo-EM structures. They also show how some loss-of-function mutant receptors elicit conformational changes (changes in fluorescence) after partial agonist binding but failed to produce measurable ionic currents, pointing to intermediate states that are stabilized in these conditions. The four fluorescence sensors developed in this study may be good tools for further studies on characterizing drugs targeting the 5-HT3R.Weaknesses:Although the major conclusions of the manuscript seem well justified, some of the comparison with the structural data may be vague. The claim that monitoring these silent conformational changes can offer insights into the allosteric mechanisms contributing to signal transduction is not unique to this study and has been previously demonstrated by using similar techniques with other ion channels.

The referee emphasizes that “some of the comparison with the structural data may be vague”. To better illustrate the structural reorganizations seen in the cryo-EM structures and that are used for VCF data interpretation, we added a new supplementary figure 3. It shows a superimposition of Apo, setron and 5-HT bond structures, with reorganization of loop C and Cys-loop consistent with VCF data.

**Reviewer #2 (Public Review):**
Summary:This study focuses on the 5-HT3 serotonin receptor, a pentameric ligand-gated ion channel important in chemical neurotransmission. There are many cryo-EM structures of this receptor with diverse ligands bound, however assignment of functional states to the structures remains incomplete. The team applies voltage-clamp fluorometry to measure, at once, both changes in ion channel activity, and changes in fluorescence. Four cysteine mutants were selected for fluorophore labeling, two near the neurotransmitter site, one in the ECD vestibule, and one at the ECD-TMD junction. Agonists, partial agonists, and antagonists were all found to yield similar changes in fluorescence, a proxy for conformational change, near the neurotransmitter site. The strength of the agonist correlated to a degree with propagation of this fluorescence change beyond the local site of neurotransmitter binding. Antagonists failed to elicit a change in fluorescence in the vestibular the ECD-TMD junction sites. The VCF results further turned up evidence supporting intermediate (likely pre-active) states.Strengths:The experiments appear rigorous, the problem the team tackles is timely and important, the writing and the figures are for the most part very clear. We sorely need approaches orthogonal to structural biology to annotate conformational states and observe conformational transitions in real membranes- this approach, and this study, get right to the heart of what is missing.Weaknesses:The weaknesses in the study itself are overall minor, I only suggest improvements geared toward clarity. What we are still missing is application of an approach like this to annotate the conformation of the part of the receptor buried in the membrane; there is important debate about which structure represents which state, and that is not addressed in the current study.
**Reviewer #3 (Public Review):**
Summary:The authors have examined the 5-HT3 receptor using voltage clamp fluorometry, which enables them to detect structural changes at the same time as the state of receptor activation. These are ensemble measurements, but they enable a picture of the action of different agonists and antagonists to be built up.Strengths:The combination of rigorously tested fluorescence reporters with oocyte electrophysiology is a solid development for this receptor class.Weaknesses:The interpretation of the data is solid but relevant foundational work is ignored. Although the data represent a new way of examining the 5-HT3 receptor, nothing that is found is original in the context of the superfamily. Quantitative information is discussed but not presented.
**Recommendations for the authors:**

**Reviewer #1 (Recommendations For The Authors):**
Here are some suggestions that may help to improve the manuscript:Page 6, point (2), typo: "L131W is positioned more profound in each ECD, its side chain (...)"

“profound” have been corrected into “profoundly”

Fig 1C: Why not compare 5-HT responses for the four sensors studied? If the reason is the low currents elicited by 5-HT on I160C/Y207W sensor, could you comment on this effect that is not observed for the other full agonist tested (mCPBG)?

The point of this figure (Fig 1G) is to show currents that desensitize to follow the evolution of the fluorescence signal during desensitization, that’s why for the I160C/Y207W sensor where 5-HT become a partial agonist we have judge more appropriate to use mCPBG acting as a more potent agonist to elicit currents with clear desensitization component. We have added a sentence in the legend of the figure to explain this choice more clearly.

Page 9, paragraph 2: "However, concentration-response curves on V106C/L131W show a small yet visible decorrelation of fluorescence and current (...)" Statistical analysis on EC50c and EC50f will help to see this decorrelation.

Statistical analysis (unpaired t test) has been added to figure 3 panel A.

Page 10, paragraph 1: the authors describe how "different antagonists promote different degrees of local conformational changes". Does it have any relation to the efficacy or potency of these antagonists? Is there any interpretation for this result?

Since setrons are competitive antagonists, the concept of efficacy of these molecules is unclear. Concerning potency, no correlation between affinity and fluorescence variation is observed. For instance, ondansetron and alosetron bind with similar nanomolar affinity to the 5-HT3R (Thompson & Lummis Curr Pharm Des. 2006;12(28):3615-30) but elicit different fluorescence variations on both S204C and I160C/Y207W sensors.

Fig. 1 panel A, graph to far right: axis label is cut ("current (uA)/..."). Colors of graph A - right are not clearly distinguishable e.g. cyan from green.

The fluorescent green color that describes the mutant has been changed into limon color which is more clearly distinguishable from cyan.

Why is R219C/F142W not selected in the study? Are the signals comparable to the chosen R219C/F142W?

We have chosen not to select R219C/F142W because the current elicited by this construct was lower than the current elicited by the construct R219C/Y140W. Moreover, the residue F142 belongs to the FPF motif from Cys-loop that is essential for gating (Polovinkin et al, 2018, Nature).

Fig. 1 legend typo: "mutated in tryptophan”

“in” has been changed by “into”

Fig. 2: yellow color (graphs in panel B) is very hard to read.

Yellow color has been darkened to yellow/brown to allow easy reading.

Fig. 4 is too descriptive and undermines the information of the study. It could be improved e.g. by representing specific structures or partial structures involved. As an additional minor comment, some colors in the figure are hard to differentiate, e.g. magenta and purple.

We have added relevant specific structures involved, namely loop C, the Cys-loop and pre-M1 loop to clarify. The intensity of magenta and purple has been increased to help differentiate the two sensor positions.

Fig S1C: it is confusing to see the same color pattern for the single mutants without the W. I would recommend to label each trace to make it clearer.

Labelling of the traces corresponding to the single mutants has been added.

Fig S2: Indicating the statistical significance in the graph for the mutants with different desensitization properties compared to the WT receptor will help its interpretation.

The statistical significance of the difference in the desensitization properties has been added to Figure S2.

**Reviewer #2 (Recommendations For The Authors):**
Overall comments for the authors:Selection of cysteine mutants and engineered Trp sites is clear and logical. VCF approach with controls for comparing the functionality of WT vs. mutants, and labeled with unlabeled receptor, is well explained and satisfying. The finding that desensitization involves little change in ECD conformation makes sense. It is somewhat surprising, at least superficially, to find that competitive antagonists promote changes in fluorescence in the same 'direction' and amplitude as strong agonists, however, this is indeed consistent with the structural biology, and with findings from other groups testing different labeling sites. Importantly, the team finds that antagonist-binding changes in deltaF do not spread beyond the region near the neurotransmitter site. The finding that most labeling sites in the ECD, in particular those not in/near the neurotransmitter site, fail to report measurable fluorescence changes, is noteworthy. It contrasts with findings in GlyR, as noted by the authors, and supports a mechanism where most of each subunit's ECD behaves as a rigid body.Specific questions/comments:I am confused about the sensor current kinetics. Results section (2) states that all sensors share the same current desensitization kinetics, while Results section (5) states that the ECD-TMD site and the vestibule site sensors exhibit faster desensitization. SF1C, right-most panel of R219C suggests the mutation and/or labeling here dramatically changes apparent activation and deactivation rates measured by TEVC. Both activation and deactivation upon washout appear faster in this one example. Data for desensitization are not shown here but are shown in aggregate in earlier panels. It is a bit surprising that activation and deactivation would both change but no effect on desensitization. Indeed, it looks like, in Fig. 1G, that desensitization rate is not consistent across all constructs. Can you please confirm/clarify?

TEVC and VCF recordings in this study show a significant variability concerning both the apparent desensitization and desactivation kinetics. This is illustrated concerning desensitization in TEVC experiments in figure S2, where the remaining currents after 45 secondes of 5-HT perfusion and the rate constants of desensitization are measured on different oocytes from different batches. Therefore, the differences in desensitization kinetics shown in fig 1.G are not significant, the aim of the figure being solely to illustrate that no variation of fluorescence is observed during the desensitization phase. A sentence in the legend of fig 1.G has been added to precise this point. We also revised the first paragraph of result section 5, clearly stating that the slight tendency of faster desensitization of V106C/L131W and R219C/Y140W sensors is not significant.

An alternative to the conclusion-like title of Results section (2) is that the ECD (and its labels) does not undergo notable conformational changes between activated and desensitized states.

This is a good point and we have added a sentence at the end of results section 2 to present this idea.

I find the discussion paragraph on partial agonist mechanisms, starting with "However," to be particularly important but at times hard to follow. Please try to revise for clarity. I am particularly excited to understand how we can understand/improve assignments of cryo-EM structures using the VCF (or other) approaches. As examples of where I struggled, near the top of p. 11, related to the partial agonist discussion, there is an assumption about the pore being either activated, or resting. Is it not also possible that partial agonists could stabilize a desensitized state of the pore? Strictly speaking, the labeling sites and current measurements do not distinguish between pre-active resting and desensitized channel conformations/states. However, the cryo-EM structures can likely help fill in the missing information there- with all the normal caveats. Please try to explicitly distinguish between a closed pore and a resting or desensitized state of the pore, to help in clarity.

We have revised the section, and hope it is clearer now. We notably state more explicitly the argument for annotation of partial agonist bound closed structures as pre-active, mainly from kinetic consideration of VCF experiments. We also mention and cite a paper by the Chakrapani group published the 4th of January 2024 (Felt et al, Nature Communication), where they present the structures of the m5HT3AR bound to partial agonists, with a set of conformations fully consistent with our VCF data.

This statement likely needs references: "...indirect experiments of substituted cysteine accessibility method (SCAM) and VCF experiments suggested that desensitization involves weak reorganizations of the upper part of the channel that holds the activation gate, arguing for the former hypothesis."

Reference Polovinkin et al, Nature, 2018, has been added.

I respectfully suggest toning down this language a little bit: "VCF allowed to characterize at an unprecedented resolution the mechanisms of action of allosteric effectors and allosteric mutations, to identify new intermediate conformations and to propose a structure-based functional annotation of known high-resolution structures." This VCF stands strongly without unclear claims about unprecedented resolution. What impresses me most are the findings distinguishing how agonists/partial agonists/antagonists share a conserved action in one area and not in another, the observations consistent with intermediate states, and the efforts to integrate these simultaneous current and conformation measurements with the intimidating array of EM structures.

We thank the referee for his positive comments. We have removed “unprecedented resolution” and revised the sentences.

It is beyond the scope of the current study, but I am curious what the authors think the hurdles will be to tracking conformation of the pore domain- an area where non-cryo-EM based conformational measurements are sorely needed to help annotate the EM structures.

We fully agree with the referee that structures of the TMD are very divergent between the various conditions depending on the membrane surrogate. We are at the moment working on this region by VCF, incorporating the fluorescent unnatural amino acid ANAP.

Minor:(1) P. 5, m5-HT3R: Please clarify that this refers to the mouse receptor, if that is correct.

OK, “mouse” has been added.

(2) Fig. 1D, I suggest moving the 180-degree arrow to the right so it is below but between the two exterior and vestibular views.

Ok, it has been done.

(3) Please add a standard 2D chemical structure of MTS-TAMRA, and TAMRA attached to a cysteine, to Fig 1.

A standard chemical structure has been added for the two isomers of MTS-TAMRA.

(4) Please label subpanels in Fig. 1G with the identity of the label site.

The subpanels have been labelled.

**Reviewer #3 (Recommendations For The Authors):**
This is solid work but I mainly have suggestions about placing it in context.(1) Abstract "Data show that strong agonists promote a concerted motion of all sensors during activation, "The concept of sensors here is the fluorescent labels? I did not find this meaningful until I read the significance statement.

We have specified “fluorescently-labelled” before sensors in the abstract.

(2) p4 "each subunit in the 5-HT3A pentamer...." this description would be identical for any pentameric LGIC so the authors should beware of a misleading specificity. This goes for other phrases in this paragraph. However, the summary of the 5HT specific results is very good.

About the description of the structure, we added “The 5-HT3AR displays a typical pLGIC structure, where….”.

(3) This paper is very nicely put together and generally explains itself well. The work is rigorous and comprehensive. But the meaning of quenching (by local Trp) seems straightforward, but it is not made explicit in the paper. Why doesn't simple labelling (single Cys) at this site work? And can we have a more direct demonstration of the advantage of including the Trp (not in the supplementary figure?) All this information is condensed into the first part of figure 1 (the graph in Figure 1A). Figure 1 could be split and the principle of the introduced quenching could be more clearly shown

detailed in a few more sentences the principle of the TrIQ approach. In addition,to be more explicit, the significative differences of fluorescence comparing sensors with and without tryptophan have been added in Figure 1, panel screening and a sentence have been added in the legend of this figure.

(4) p10 "VCF measurements are also remarkably coherent with the atomic structures showing an open pore (so called F, State 2 and 5-HT asymmetric states), "This statement is intriguing. What do these names or concepts represent? Are they all the same thing? Where do the names come from? What is meant here? Three different concepts, all consistent? Or three names for the same concept?

We have tried to clarify the statement by making reference to the PDB of the structures.

(5) "Fluorescence and VCF studies identified similar intermediate conformations for nAChRs, ⍺1-GlyRs and the bacterial homolog GLIC(21,32-35). "Whilst this is true, the motivation for such ideas came from earlier work identifying intermediates from electrophysiology alone (such as the flip state (Burzomato et al 2004), the priming state (Mukhatsimova 2009) and the conformational wave in ACh channels grosman et al 2000). It would be appropriate to mention some of this earlier work.

We have incorporated and described these references in the discussion. Of note, we fully quoted these references in our previous papers on the subject (Menny 2017, Lefebvre 2021, Shi 2023), but the referee is right in asking to quote them again.

(6) "A key finding of the study is the identification of pre-active intermediates that are favored upon binding of partial agonists and/or in the presence of loss-of-function mutations. "Even more fundamental, the idea of a two-state equilibrium for neurotransmitter receptors was discarded in 1957 according to the action of partial agonists.DEL CASTILLO J, KATZ B (1957) Interaction at end-plate receptors between different choline derivatives. Proc R Soc Lond B Biol SciSo to discover this "intermediate" - that is, bound but minimal activity - in the present context seems a bit much. It is a big positive of this paper that the results are congruent with our expectations, but I cannot see value in posing the results as an extension of the 2-state equilibrium (for which there are anyway other objections).As for intermediates being favoured by loss of function mutations, this concept is already well established in glycine receptors (Plested et al 2007, Lape et al 2012) and doubtless in other cases too.I do get the point that the authors want to establish a basis in 5-HT3 receptors, but these previous works suggest the results are somewhat expected. This should be commented on.

We also agree. We replace “key finding” by “key observation”, quote most of the references proposed, and explicitly conclude that “The present work thus extends this idea to the 5HT3AR, together with providing structural blueprints for cryo-EM structure annotation”.

(7) "In addition, VCF data allow a quantitative estimate of the complex allosteric action of partial agonists, that do not exclusively stabilize the active state and document the detailed phenotypes of various allosteric mutations."Where is this provided? If the authors are not motivated to do this, I have some doubts that others will step in. If it is not worth doing, it's probably not worth mentioning either.

Language has been toned down by “In addition, VCF data give insights in the action of partial agonists, that do not exclusively stabilize the active state and document the phenotypes of various allosteric mutations."

(8) Figure 1G please mark which construct is which.

This has been added into Figure 1G